# Adaptive Attention Link-based Regularization for Vision Transformers

## Abstract

Although transformer networks are recently employed in the various vision tasks with the outperforming performance, large training data and a lengthy training time are required to train a model to disregard an inductive bias. Using trainable links between the channel-wise spatial attention of a pre-trained Convolutional Neural Network (CNN) and the attention head of Vision Transformers (ViT), we present a regularization technique to improve the training efficiency of ViT. The trainable links are referred to as the attention augmentation module, which is trained simultaneously with ViT, boosting the training of ViT and allowing it to avoid the overfitting issue caused by a lack of data. From the trained attention augmentation module, we can extract the relevant relationship between each CNN activation map and each ViT attention head, and based on this, we also propose an advanced attention augmentation module. Consequently, even with a small amount of data, the suggested method considerably improves the performance of ViT while achieving faster convergence during training.

## 1 Introduction

Convolutional Neural Networks (CNN) have become standard for solving image-related tasks using deep neural networks since the advent of large publicly available datasets [1, 2]. Recently, models with attention mechanisms mainly adopted in the area of natural language processing are becoming to take part in solving image-based tasks, which is called the Vision Transformer (ViT) [3, 4]. ViT is a transformer-based neural network fed by the patches of images with class-token for classification, replacing its input of the embedded words in natural language processing. Although ViT shows impressive accuracy compared to modern CNNs by ignoring the inductive bias of locality, a significant amount of data is required to train the model to achieve satisfactory performance without overfitting issues. Furthermore, most researchers with the limited computing hardware are not affordable to train the ViT due to its lengthy training time.

The overfitting and lengthy training issues must be solved to broaden the usability of ViT, so many recent studies have tried to solve the problem [5, 6, 7, 8, 9, 10]. We can divide the studies by three categories: the advanced architecture-based method [4, 8, 11, 12], the parameter compression-based method [13, 14], and the knowledge distillation-based method [6, 15]. The advanced architecture-based methods manipulate the architecture of ViT to achieve improved training efficiency and generalized prediction even with the small dataset. On the other hand, the parameter compression-based methods focus on a low-rank approximation of the transformer encoder in ViT, which results in the boosted training speed and the suppressed overfitting issue. The knowledge distillation-based methods utilize the prediction of additional CNN models to avoid the overfitting problem and achieve rapid training convergence. The previous studies have shown the meaningful development of ViT for the small dataset and the reduced training time.

However, the previous studies have the remaining limitations where the training datasets must be equivalent for both the student and teacher models. The architectural manipulation of initial ViT [3] cannot be applied to different versions of ViTs, hence limiting the handling of new ViT models. Even though the knowledge distillation-based methods can be employed with the small manipulation of a model such as the knowledge distillation token, it should be assured to have the initial models trained by a target dataset, which takes the additional costs for the acquisition of initial models before the training of main ViT model.

In this paper, we propose a novel regularization method to reduce the convergence time and avoid the overfitting problem on a small dataset, simultaneously. The proposed method utilizes an *attention augmentation module* containing multiple trainable weights that estimate the affinity between the channel-wise activation map of CNN and the head-wise attention map of ViT. Since the attention augmentation module is located to regularize the attention of ViT heads, the architecture of ViT can be perfectly preserved, which lets us enable to employ the proposed algorithm in ViT variants based on the attention maps. In addition, since the attention map can be obtained even when the task of the pre-trained CNN is not equivalent to the target task, we can employ our method without the pre-trained CNN model with the same target dataset. We validate our regularization method by using ImageNet and CIFAR10 datasets with various scenarios, which show the outperforming accuracy and the reduction of epochs required for its training convergence. Furthermore, we investigate the important factors for ViT to avoid the overfitting issue by analyzing the trained weights of the attention augmentation module, and through the investigation, we present the dissimilarity of the deep layers' roles between CNN and ViT.

We can summarize our contributions as following:

- We propose a novel regularization method to resolve the issues of overfitting and lengthy training time of ViT through the trainable attention links between the ViT attention maps and CNN activation maps.
- The proposed scheme preserves the original architecture of ViT, which results in its general employment regardless of the architecture of ViT.
- Through the proposed algorithm, the performance of ViT can be dramatically improved with the limited size of dataset, and the training time is reduced without the loss of performance in various scenarios.
- The relationship between ViT and CNN is analyzed in terms of attentional regions, which validates the analysis from the previous studies.

## 2 Related Work

### 2.1 Transformers in Vision

Transformer models introduced by [16] are neural networks that purely utilize the attention mechanism. While they have been used broadly in the field of natural language processing, Vision Transformer (ViT) [3] adapted them in the domain of computer vision with minimal modification to its architecture. ViT showed comparable performance to CNN in the condition of large pre-training. For the advanced optimization of ViT, CaiT [4] used layer normalization in ViT layers and changed the input location of class token to prevent saturation of performance in deep layers. Swin Transformers [12] adopted a hierarchical transformer that computes shifted windows to make it suitable for the vision domain. PiT [10] introduced the concept of pooling in ViT from CNN, improving the generalization of ViT. T2T-ViT [8] enhanced sample efficiency by reshaping input tokens and changing the backbone of networks motivated by several CNN architectures. Raghu et al. [17] measured the similarity of representations between specific layers of CNN and ViT using centered kernel alignment. With additional relative positional encoding, Cordonnier et al. [18] proved attention mechanisms in ViTs can perform as convolution layers in CNNs and showed their functional similarity. From the investigation, ConViT [7] was motivated to use relative positional encoding to give locality – the inductive bias of CNNs – to ViT.

The research was extended to [19], reparameterizing pre-trained convolutional layers as a format of ConViT. Refiner [11] tackled the over-smoothing problem between tokens in deep layers of ViT, and relieve it by projecting attention heads into the higher dimensions and applying convolution directly to attention maps to learn local relationship among the tokens. Those variants of ViT improved the optimization and data efficiency of the initial ViT model by modifying the architecture itself.

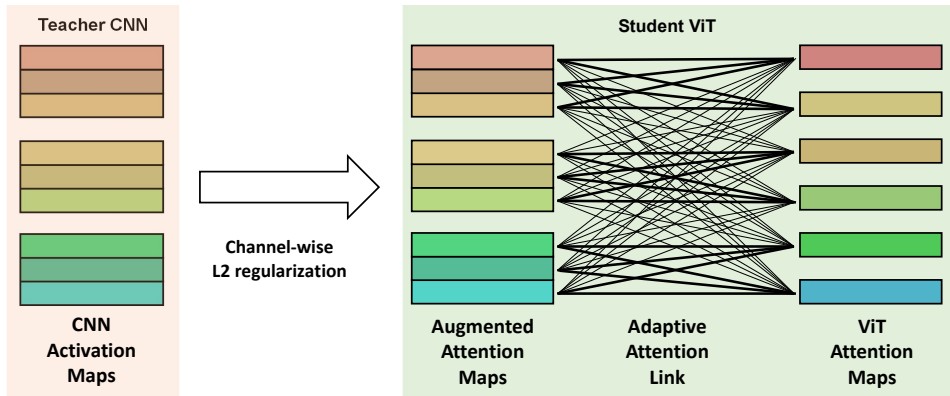

Figure 1: Our distillation procedure. ViT attention maps are augmented through adaptive attention links, which build linear combinations of different original maps. Then, those augmented attention maps mimic CNN activation maps at one-on-one correspondence.

However, our method does not touch any part of ViT modules, but only connects attention between ViT and CNN, transferring attention through links to give ViT a learning signal from the teacher.

## 2.2 Knowledge Distillation:

In knowledge distillation, a student model leverages a pre-trained teacher model's soft prediction divided by the same temperature values [20]. The softened predictions can be regarded as label-smoothing, and by using them, the student model can achieve the data augmentation effect. Distillation between different types of neural architectures has also been proposed, DistilBERT [21] showed the effectiveness of a distilled knowledge from BERT [22] into LSTM [23]. DeiT [6] distilled knowledge from CNN to ViT, which seems similar to our work. The knowledge distillation has been also employed into the transformer-based model of natural language processing, which results in the performance improvement by using the teacher model [24, 25] However, in contrast to the previous study using the prediction for the knowledge distillation, our framework transfers the knowledge based on the similarity of the latent feature maps. As a result, we can extend the range of teacher models to cover the models in which the prediction vectors differ from the prediction of the task.

On the other hand, we can transfer latent representations of teacher models to those of student models. FitNets [26] improved the stability of deep network training by guiding the latent layers to the teacher's well-trained latent representation. Zagoruyko et al. [27] considered attention as projected activation maps of CNN into a spatial dimension, which could be regarded as spatial attention. They showed that spatial attention contains valuable information that is useful to improve the performance of the student network. Kim et al. [28] used a paraphraser to extract and pass the teacher's knowledge to the student's translator to learn its representation. Meanwhile, Heo et al. [5] demonstrated that the knowledge transfer based on the neurons' activation is a more classification-friendly approach than the direct transfer using output values. Attention-based feature distillation [29] measured the similarities between teacher and student features through attention, which determines the importance of knowledge to transfer.

## 3 Attention Link-based ViT Regularization

In this section, we first explain the backgrounds of the self-attention mechanism and ViT. Then, we explain the method to extract the attention maps from ViT, followed by the description of the architecture and the training method of the augmented attention module is described. The overall framework is depicted in Fig. 1

### 3.1 Background of ViT

We first explain the self-attention mechanism and the original ViT model referred to by [3]. The self-attention mechanism mimics the human cognition system making the attention to the external

stimulus, which is designed by a transformer-based model with the attention matrix estimated by pairs of key and query.

**Self-attention Mechanism:** We define the input sequence by $\mathbf{X} \in \mathcal{R}^{L \times D_{in}}$ where $L$ is the length of the sequence and $D_{in}$ means the dimension of one sequential element in the sequence. Then, we can estimate the elements of the attention mechanism composed of key, query, and value vectors by linearly projecting the input sequence by the corresponding embedding weights $\mathbf{W}_k$, $\mathbf{W}_q$, and $\mathbf{W}_v$, respectively. Thus, when we define the key, query, and value vectors by $\mathbf{Q}$, $\mathbf{K}$, and $\mathbf{V}$, respectively, we obtain the vectors as following:

$$\mathbf{K} = \mathbf{X}\mathbf{W}_k, \mathbf{Q} = \mathbf{X}\mathbf{W}_q, \mathbf{V} = \mathbf{X}\mathbf{W}_v, \tag{1}$$

where $\mathbf{W} \in \mathcal{R}^{D_{in}} \times D_{head}$ from $\mathbf{W} \in \{\mathbf{W}_k, \mathbf{W}_q, \mathbf{W}_v\}$ and $D_{head}$ is the dimension of the head embedding.

Then, the self-attention of the head can be estimated by:

$$f(\mathbf{X}) = s\Big(\mathbf{Q}\mathbf{K}^T/\sqrt{D_{head}}\Big)\mathbf{V} \in \mathcal{R}^{L \times D_{head}}, \tag{2}$$

where $s(\mathbf{Z})$ is a function to transfer each row vector of input matrix $\mathbf{Z}$ by softmax. According to the derivation, self-attention can consider the semantic dependency among sequential inputs.

Many transformer-based models are based on the architecture stacked by the Multi-Head Self-Attention layers (MHSA) containing multiple self-attention heads with independent embedding weights. For the given input $\mathbf{X}$, we define the output of $m$-th self-attention head at $n$-th level depth by $f_{(m,n)}(\mathbf{X})$. Then, we denote the corresponding key, query, and value vectors by $\mathbf{K}_{(m,n)}$, $\mathbf{Q}_{(m,n)}$, and $\mathbf{V}_{(m,n)}$, respectively.

**ViT Framework:** The original ViT model directly employed the conventional transformer-based model built for the natural language processing of the visual classification task. We can summarize the inference process of the original ViT as following. At first, we divide an input image by $P^2$ patches with the same size and sequentially order the patches after their vectorization. Since the transformer network is invariant to the order of the sequential data, ViT concatenates positional embedding vectors to the input patches to represent the original position of the patch.

We define the sequential data obtained from one image by $\mathbf{X}_0 \in \mathcal{R}^{P^2 \times D_{im}}$, where $D_{im}$ is the size of the vector linearly projected from the vectorized image patch and the positional embedding vector. Before we feed $\mathbf{X}_0$ into the transformer modules, the trainable class token sized by $\mathcal{R}^{D_{im}}$ is sequentially connected ahead of $\mathbf{X}'_0$, which results in $\mathbf{X}_0 \in \mathcal{R}^{(P^2+1) \times D_{im}}$.

The transformer-based encoders of ViT are the modules containing the series of a layer normalization, a self-attention multi-head module, a fully connected layer, and a layer normalization, where every normalization layer has a residual connection. We define the serial process by a function of $\mathbf{X}_{n+1} = g(\mathbf{X}_n) \in \mathcal{R}^{(P^2+1) \times D}$ where $D$ is the size of latent vectors. When $N$ modules are stacked in the transformer-based encoder, the class-wise score is estimated by linearly projecting the final output of the class token as following: $p(cls|\mathbf{X}) = softmax(FC(\mathbf{X}_N))$. For a detailed explanation of ViT, you can be referred to [3].

### 3.2 Attention Map Extraction

We need to compare the ViT attention map and the CNN activation map for our regularization-based algorithm. Instead of the relative positional embedding [7] or the attention bias [11], we preserve the original architecture of ViT to generalize the usability of our framework to cover the ViT variants.

To obtain the attention map from the original ViT, we utilize the attention value between the class token and the image patch. The class token takes a key role to determine the final prediction, so we can assume that the attention to the class token may represent the importance of image patches for the classification result. Thus, when feeding the class token into the transformer module as its query vector, we obtain the attention value by estimating the dot product between the embedding vectors of the class token and the corresponding image patch. Then, for $m$-th head in $n$-th multi-head self-attention layer, we can estimate the attention value as:

$$\mathbf{A}_{(m,n)} = Rec\Big(s\big(\mathbf{Q}_{(m,n)}[0]\mathbf{K}_{(m,n)}[1:]^T\big)\Big) \in \mathcal{R}^{P \times P}, \tag{3}$$

where $Rec$ is a function to reconstruct the rectangular matrix of $\mathcal{P} \times \mathcal{P}$ from its input vector of $\mathcal{P}^\in$ according to the order of the sequential patches, and $\mathbf{Q}_{(m,n)}[0]$ and $\mathbf{K}_{(m,n)}[1:]$ represent the first query vector of the class token and the key vectors of the image patches, respectively.

In the case of the CNN activation map, we extract the activation maps after the normalization of every convolution block. Instead of integrating the channel-wise activation maps, we consider the separated activation maps independently to improve the degree of freedom of our attention augmentation module. In contrast to the constant resolution of ViT attention maps, the resolution of the CNN activation maps decreases with deep layers by pooling layers and strides of convolution layers. Thus, to preserve the resolution of every activation map, we resize all the CNN activation maps to have the same size with the ViT attention maps by using bicubic interpolation. We define the $c$-th resized CNN activation map by $\mathbf{B}_c$, where $c \in \{1, ..., C\}$ and $C$ is the number of entire CNN activation maps.

### 3.3 Attention Augmentation Module

#### 3.3.1 Module Architecture

Even though both the CNN activation and ViT attention maps represent the key parts of the target object for the prediction, their distribution such as a center point and a variance would be different from each other due to the dissimilarities of their operations. For example, while the ViT attention map is distributed between 0 and 1 because of the softmax estimation, the values in the CNN activation map are normalized by a batch normalization, which can contain negative values. Furthermore, in general, the number of CNN activation maps is much larger than the number of ViT attention maps due to the large channel-wise depth of CNNs. As a result, it is impossible to directly compare each of the CNN activation maps with the ViT attention maps.

The attention augmentation module is designed to solve the problems of different distributions and a varying number of maps. We design the attention augmentation module to contain multiple attention links which are the trainable weight parameters to scale the ViT attention maps. By estimating the weighted summation of ViT attention maps with the attention links, we can obtain the augmented attention maps where the number is equivalent to the number of CNN activation maps. Thus, we can estimate the augmented attention maps as following:

$$\mathbf{A}_c^+ = \sum_{m,n=1}^{M,N} w_c^{(m,n)} \mathbf{A}_{(m,n)} + b_c, \tag{4}$$

where $w_c^{(m,n)}$ and $b_c$ are the attention link and a trainable bias for $c$-th augmented attention map ($c \in \{1, ..., C\}$), respectively. $M$ is the number of self-attention heads in one level depth and $N$ presents the maximum level depth. Note that the weight of attention link $w_c^{(m,n)}$ is used to analyze the strength of connectivity for each CNN/ViT layer in section 4.1.

We implement the augmented attention module by a $1 \times 1$ convolution layer generating $C$ augmented attention maps from a tensor of $\mathcal{R}^{P^2 \times MN}$ where the ViT attention maps are stacked. Because we only use the augmented attention maps only for the training loss, the attention augmentation module has no role in the inference, which can be removed after the training of ViT.

#### 3.3.2 Module Training

By using the augmented attention module, we can obtain the same number of augmented attention maps $\mathbf{A}_c^+$ with the CNN attention maps $\mathbf{B}_c$. To ignore the remaining scale gap between the two maps, we first apply the $l2$ normalization, and then the mean squared error is estimated to build the attention-based regularization loss as:

$$\mathcal{L}_{att} = \left\| \mathbf{A}_c^+ / \|\mathbf{A}_c^+\|_2 - \mathbf{B}_c / \|\mathbf{B}_c\|_2 \right\|_2. \tag{5}$$

Then, we integrate the attention-based regularization loss $\mathcal{L}_{att}$ with the cross-entropy loss $\mathcal{L}_{CE}$ of original ViT as:

$$\mathcal{L} = \mathcal{L}_{CE} + \lambda \mathcal{L}_{att}, \tag{6}$$

where $\lambda$ is a scaling factor to control the effect of our regularization. Since the regularization loss can work as an obstacle to ignoring the inductive bias, referred by [7], we suppress the value of $\lambda$ at the specified epochs to increase the effectiveness of the cross-entropy loss. We exponentially decay the value of $\lambda$ by multiplying a decay constant between 1 and 0 at every epoch.

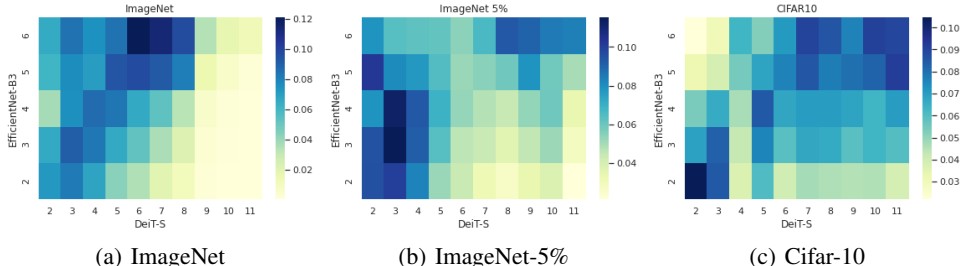

| (a) ImageNet | (b) ImageNet-5% | (c) Cifar-10 |

Figure 2: Relations between CNN Activation Maps and ViT attention maps. The x-axis and y-axis indicates the depth of ViT layer and CNN block, respectively. We obtain the heatmap by averaging the magnitudes of attention links.

| CNN Block Level | | 1 | 2 | 3 | 4 | 5 | 6 | 7 |
|---|---|---|---|---|---|---|---|---|
| ViT Layer Level | $\alpha$-link | $1 \sim 3$ | $1 \sim 5$ | $3 \sim 5$ | $4 \sim 6$ | $4 \sim 6$ | $6 \sim 11$ | $7 \sim 12$ |
| | $\beta$-link | $1 \sim 2$ | $1 \sim 5$ | $3 \sim 5$ | $4 \sim 6$ | $4 \sim 7$ | $6 \sim 10$ | $7 \sim 10$ |

Table 1: Selective Link Configuration.

## 4   Link Selection for Advanced Regularization

In this section, we build the advanced architecture of the attention augmentation module based on the analysis of the fully-trained attention links. After showing the resultant attention links, we explain the advanced link designed by considering the relation between CNN activation and ViT attention maps.

### 4.1   Analysis of Resultant Attention Links

We visualize heat maps to show the scale distribution of the attention links after their training. To compare the relationship between the ViT attention and the CNN activation maps, we only consider the magnitude of the weight parameters in the attention links. As shown in Fig. 2, we obtain multiple heat maps by using three datasets including *ImageNet* [30], a 5% subset of *ImageNet*, and *Cifar-10* [31].

As analyzed in many previous studies [18, 17], the ViT attention maps are highly related to the CNN activation maps located at a similar level. The results validate that the self-attention multi-heads of ViT can train the hierarchical information by the stacked architecture, which is similar to the training mechanism of CNN. Thus, the CNN activation maps would be helpful to regularize the ViT attention maps if we can find their level similarity.

In addition to the well-known hypothesis, the heat maps from the attention links show the interesting characteristic where the high-level heads are not regularized when the large dataset is given. When a large dataset is provided, we can observe the suppressed magnitudes of the attention links at high-level heads in comparison to small datasets. The discovery exploits the high-level heads should not be trained by the regularization of the high-level CNN layers, which verifies the high-level heads can represent more complicated semantic information than CNN layers. In other words, the representation can be seen out of the inductive bias of CNNs, so we can show semantic information that overwhelms the inductive bias is hard to be trained without a large dataset.

DeiT [6] and Swin Transformers [12] showed that the employment of the inductive bias of CNN is effective for the training of ViT. At the end of the training, we observed that high-level heads are disconnected from augmented attention maps, which means they are no more regularized by CNN activation maps. This indicates that high-level heads would escape the inductive bias of locality, and they are trained by a long-range dependency that cannot be acquired by CNNs.

We can summarize the two hypotheses from the analysis as following:

- The ViT attention and the CNN activation maps are similar to each other at a similar level.

- The high-level ViT heads can present the semantic information that cannot be represented by the CNN layers, but training the semantic information requires a large dataset.

| Train | Top-1 | | | | Top-5 | | | |
|---|---|---|---|---|---|---|---|---|
| Size | DeiT | ConViT | AAL (Ours) | Gap | Deit | ConViT | AAL(Ours) | Gap |
| 5% | 34.8% | 47.8% | **51.7%** | 49%/8% | 57.8% | 70.7% | **75.9%** | 31%/7% |
| 10% | 48.0% | 59.6% | **64.7%** | 35%/9% | 71.5% | 80.3% | **85.8%** | 20%/7% |
| 30% | 66.1% | 73.7% | **76.1%** | 15%/4% | 86.0% | 90.7% | **93.0%** | 8%/3% |
| 50% | 74.6% | 78.2% | **78.9%** | 6%/1% | 91.8% | 93.8% | **94.5%** | 3%/1% |
| 100% | 79.9% | **81.4%** | 81.0% | 1%/0% | 95.0% | **95.8%** | 95.5% | 1%/0% |

Table 2: ImageNet test accuracy with different sampling ratios. The Gap columns represent the relative performance improvement of the AAL over DeiT and ConViT, respectively.

## 4.2 Selective Attention Link

Based on the analysis, we additionally propose the selective attention link to improve the training efficiency of the attention augmentation module. Instead of the full link in the original attention augmentation module, only a part of the attention links are utilized to obtain the augmented attention maps. Based on the two hypothesis from our analysis, the augmented attention map is generated only by the ViT attention maps with the similar levels, and no link is connected to the high-level ViT attention maps when the training dataset is sufficiently large.

Accordingly, we build two types of selective attention link, which are denoted by $\alpha$-link and $\beta$-link. $\alpha$-link connects the ViT attention maps to only the augmented attention maps at a similar level. $\beta$-link is similar to the $\alpha$-link but the links to the high-level heads are entirely disconnected. The detailed connections are given in Table 1.

## 5 Experiments

In experiments, we showed that transferring attention from pre-trained CNN models to ViTs can inject CNN's inductive bias (i.e locality) naturally in standard self-attention layers, without the necessity of additional modules extending the self-attention network. Thus, we examine how efficiently the method helps ViT to be converged for optimal performance, especially showing a large gap in a small data regime.

### 5.1 Experimental Settings

**Implementation Details:** The computing resource used in our experiments is Nvidia A100. If not mentioned otherwise, the student ViT model used for experiments is DeiT-S (distilled version) and used EfficientNet-B3 [32] as the

| Models | DeiT-B | ConViT | AAL |
|---|---|---|---|
| Top-1 | 97.5% | 95.4% | 97.5% |

Table 3: CIFAR10 Top-1 test accuracy

teacher CNN model. We set $\lambda$ to 2000 and the decay constant for $\lambda$ is set to 0.99 for the first 200 epochs and 0.98 for the last 100 epochs. For a fair comparison, we preserve the values of the remaining hyperparameters and the training strategies from our baseline model of DeiT [6].

**Comparisons and Dataset:** For comparison, we consider two previous studies, which include DeiT and ConViT. DeiT utilizes the knowledge distillation method to improve the ViT-based models, and ConViT shows the state-of-the-art performance when the training data is given sufficiently even without using the knowledge distillation methods. To show the generality of our algorithm, we utilize four classification datasets: ImageNet, CIFAR10, Caltech-UCSD Birds-200-2011 (CUB-200), and Oxford 102 Flowers (Flower-102). In the case of ImageNet, we extract the subsets randomly sampled with the various ratios (5%, 10%, 30%, 50%), maintaining class balance, to show the validity of the proposed algorithm when the insufficient data is given for the training.

### 5.2 Quantitative Results

We first perform the comparisons with the various subsets of ImageNet. As shown in Table 2, the proposed algorithm shows the state-of-the-art performance when the ImageNet subsets are used to train the model. The performance of our framework is similar to that of ConViT when the entire dataset is considered for training. However, the performance gap between our framework and ConViT becomes enlarged with the insufficient training data. Furthermore, we should notice that

| Train | Top-1 | | | Top-5 | | |
|---|---|---|---|---|---|---|
| Size | Full-link | Selective-link | Gap | Full-link | Selective-link | Gap |
| 5% | 48.9% | **51.7%** | 5.7% | 73.6% | **75.9%** | 3.1% |
| 10% | 63.0% | **64.7%** | 2.6% | 84.6% | **85.8%** | 1.4% |
| 30% | 75.2% | **76.1%** | 1.2 % | 92.4% | **93.0%** | 0.6% |
| 50% | 78.5% | **78.9%** | 0.5% | 94.3% | **95.0%** | 0.7% |
| 100% | **81.0%** | 80.9% | -0.1% | 95.5% | **95.5%** | 0.0% |

Table 4: ImageNet test accuracy with different sampling ratios. The Gap columns represent the relative performance improvement of Selective-link.

| Teacher Model | Student Model | Teacher Model Top-1 | Student Model Top-1 |
|---|---|---|---|
| ResNet34 | DeiT-S w/ distill | 73.3% | 79.4% |
| EfficientNet-B3 | DeiT-B w/ distill | 81.1% | 82.8% |

Table 5: ImageNet test accuracy with various teacher and student models

EfficientNet-B3 that is our teacher model needs only 12.2M parameters, which is much smaller than 86.6M parameters of RegNetY-16GF [33] used in DeiT [6]. Thus, we can validate that our proposed framework can overwhelm DeiT-B even by using the light teacher model. In addition, while ConViT-S needs 5M more parameters than ours or DeiT, our method outperforms both of DeiT and ConViT-S, which validates the efficiency of our framework. The quantitative results for CUB and Flower datasets are represented in the supplementary material.

Table 3 shows the experimental results with CIFAR10 dataset. The results verifies that the proposed algorithm can increase the robustness to the insufficient size of training data. In addition, in the DeiT paper, 7200 training epochs were needed to achieve 97.5% top-1 test accuracy when training from scratch using the DeiT-B model which has more attention heads than DeiT-S. On the other hand, our method only needed to train 300 training epochs to reach the same test accuracy while using the DeiT-S model, which validates its training efficiency.

## 5.3 Analysis

In addition to the following analysis, we present the additional experiments to show the validity of our framework in the supplementary material. The additional experiments include the performance of weakly supervised object localization, the qualitative results for attention maps, the learning curve, and epoch-wise qualitative changes of attention links.

**Effectiveness of Selective Links:** To show that our selective attention link-based transfer efficiently matches ViT attention maps with CNN activation maps, we compared two different settings on the attention augmentation module. *Full-link* fully connects each ViT attention map to produce augmented attention maps that match CNN activation maps as one-to-one channel-wise correspondence. In the case of the full ImageNet dataset, $\beta$-link was used for the selective link, while we utilized $\alpha$-link for the other small datasets. As shown in Table 4, the attention transfer with a fully connected attention link shows superior performance to the accuracy of DeiT and ConViT (Table 2) in low data regime, and the selective attention links show further improvement from its results.

**Robustness to Variety of Models:** We add results with the variants composing of different teacher and student models to show the generality of our method upon various environments. As shown in Table 5, the proposed framework successfully improves the performance of its teacher model even with the different teacher and student models. Interestingly, when we use a light teacher model, we can achieve the large performance gap between the teacher and student models.

**Data and Model Efficiency:** Our additional trainable module, which is the attention augmentation module, includes only a single 1x1 Conv layer which augments the attention maps of the student ViT. In our default settings, the number of the parameter is 0.068M, which is quite small compared to DeiT-S of 22M parameters. As we mentioned, the lengthy training time of ViT is a critical drawback especially when the computational resources are limited. The reduced training time of our method can be validated through the learning curve represented in the supplementary material.

| Strong Data Aug. (Default) | | | | Weak Data Aug. | | |
|---|---|---|---|---|---|---|
| Methods | Top-1 | Top-5 | | Methods | Top-1 | Top-5 |
| Cross Entropy (CE) | 91.3% | 99.6% | | Cross Entropy (CE) | 84.2% | 98.7% |
| CE + AAL | 97.4% | 99.9% | | CE + AAL | 92.5% | 99.7% |
| CE + Soft Distillation | 91.0% | 99.6% | | CE + Soft Distillation | 84.0% | 98.9% |
| CE + Hard Distillation | 92.0% | 99.8% | | CE + Hard Distillation | 85.1% | 99.0% |
| CE + AAL + Hard Dist. | 96.5% | 99.9% | | CE + AAL + Hard Dist. | 94.1% | 99.7% |

Table 6: Ablation Test with different settings of data augmentation and distillation methods

**Ablation studies:** For additional verification of our knowledge transfer method, we trained on CIFAR10 with different scenarios. To sternly check the performance difference of knowledge distillation effect from each method, we used much weaker data augmentation than the setting used in other experiments with only simple techniques such as random crops and horizontal flips. This allows us to confirm the data efficiency in a low data regime. In addition, we compared our method to other knowledge distillation methods introduced by DeiT with a teacher model pre-trained on the CIFAR10 dataset. As shown in Table 6, for both soft label distillation and hard label distillation, our method outperforms the class prediction-based distillation method. From this result, we could infer that directly transferring attention gives a better learning signal than giving the teacher model's output predictions. Furthermore, we could confirm that knowledge earned from a large dataset can give a good learning direction. Compared to the result of Table 3 where the teacher model pre-trained by ImageNet was used, the performance drops due to the lack of information in the teacher model pre-trained by CIFAR10. This could be another advantage since teacher models in DeiT are restricted to be trained on the target dataset to give proper output prediction.

**Various Baseline:** In Table 7, we show that applying our method is not only limited to standard ViT. In the experiments, we employ our method to Pooling based ViT (PiT-S) [10], and we observed the sample efficiency of the model increased by a large margin using our method.

**Robustness to random initialization:** We perform several trials with different random seeds as shown in Table 8. Our algorithm shows consistency even with the various initial parameters. Due to our limited computation, we run 240 epochs of training in contrast to 300 epochs of training in the default setting.

| | Pit-S | Pit-S + AAL |
|---|---|---|
| Top-1 | 12.2% | 44.0% |
| Top-5 | 25.2% | 67.3% |

Table 7: Our method with PiT (ImageNet 5%)

**Prediction-based distillation and Fine-tuning:** Our method can be jointly applied with class prediction-based distillation. Thus, both Table 2 and Table 3 verify that our method can show synergy with the distillation method proposed in [6]. In addition, we perform the additional comparison to fine-tuning algorithms with self-supervised learning (SSL) for ViT [34], SSL with linear classifier, and SSL with k-NN classifier respectively show 77% and 74.5% for ImageNet top-1 test accuracy. Every accuracy is lower than our performance of 81.0% with the same ViT model, which validates the synergy of our method with the self-supervised fine-tuning mechanisms.

# 6 Conclusion

In this paper, we have introduced a novel method of transferring knowledge from CNN to ViT. By accessing attention of CNNs and adaptively adopting them, student ViT was able to earn high quality of learning signal with CNN's inductive bias. By applying our method, we could train ViT in less training epochs without overfitting even with the small dataset or limited labeled data. Also, we revealed relations between intermediate representations from those different types of neural networks, which varied due to the training dataset. Furthermore, by analyzing those relationship with trained attention links, we could take advantage of more efficient connection between networks. We leave wider application of our methods to new ViT architectures in future works.

| | Trial I | Trial II | Trial III |
|---|---|---|---|
| Top-1 | 47.3% | 46.5% | 47.2% |
| Top-5 | 71.9% | 71.3% | 72.0% |

Table 8: Repeated trials (ImageNet 5%)

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
