# Adaptive Attention Link-based Regularization for Vision Transformers

## Supplementary Material

## A Fine-Grained Image Classification

| | CUB-200 (AAL) | | CUB-200 (DeiT) | | Flowers-102 (AAL) | | Flowers-102 (DeiT) | |
|---|---|---|---|---|---|---|---|---|
| Methods | Top-1 | Top-5 | Top-1 | Top-5 | Top-1 | Top-5 | Top-1 | Top-5 |
| Scratch | 51.05% | 80.03% | 26.49% | 53.67% | 94.13% | 99.14% | 87.16% | 96.33% |
| Transfer | 84.76% | 96.08% | 83.87% | 95.87% | - | - | - | - |

Table 9: Classification accuracy on CUB-200-2011 and Flowers-102. 'Scratch' results are obtained by training-from-scratch, and 'Transfer' results by learning from the ImageNet pretrained model. We trained 300 epochs for both scenarios.

Since our method mainly considers in low-data regime, we regard fine-grained classification as another appropriate task to test our data efficiency. We test classification accuracy on CUB-200-2011 and Oxford flowers-102, both of which have a small number of samples per class.

## B Weakly Supervised Object Localization

| IoU threshold | DeiT | AAL |
|---|---|---|
| 0.3 | 50.3 | 64.5 |
| 0.5 | 20.0 | 30.7 |
| [0.3, 0.5] | 35.2 | 48.3 |
| [0.3, 0.5, 0.7] | 25.0 | 34.2 |

Table 10: Localization accuracy in WSOL task

We additionally evaluate our framework through Weakly Supervised Object Localization (WSOL), which is frequently used to show the space awareness. WSOL trains the network model to classify the input image and evaluates the localization of target objects. We determine the position of the target objects by averaging the entire attention maps of Eq.3. We measure the localization performance by using the Intersection of Union (IoU) in CUB-200-2011. The results are given in Table 10. We refer [*] for the evaluation method of WSOL. We use MaxBoxAcc, measuring how the box generated from the attention map overlaps with the ground truth box with a IoU threshold. While the default setting suggests is 0.5, We demonstrate results with different IoU thresholds. The result of multiple IoU threshold indicates average of scores from each threshold in a list. Compared to DeiT, although the proposed algorithm shows only 1% performance improvement, its localization accuracy is 53.5% higher than that of DeiT at default setting, which validates the space awareness of our knowledge distillation scheme.

[*] "Evaluating Weakly Supervised Object Localization Methods Right", Choe et al.

## C Qualitative Results for Attention Maps

For qualitative analysis on the effect of our regularization method to attention maps, we compare the attention on objects which are acquired by averaging the entire attention maps of Eq.3. For each models, we used same DeiT-S model as baseline but only differ in training strategy, (b) DeiT-S with hard distillation and (c) DeiT-S with AAL. For training, we used CUB-200-2011 dataset, which only labeled for classification with small number of samples. After training, we obtained class attention map by aforementioned process. We could easily tell that model trained with AAL localizes on object to classify better than DeiT, with higher intensity value on the area. From result, we could also infer that the inductive bias of CNN-locality is transferred more successfully by observing its attention on object when trained with AAL. Quantitative results on the attention maps are followed at section B.

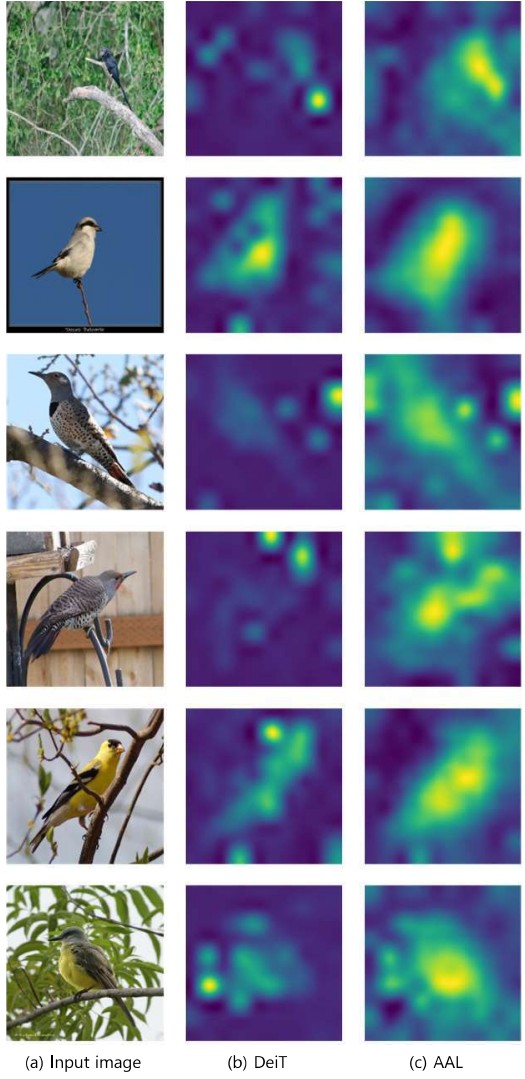

(a) Input image          (b) DeiT          (c) AAL

Figure 3: Attention maps obtained by a DeiT and AAL after 300 epochs of training on Caltech-UCSD Birds-200-2011. For each rows, we illustrate the channel-wise mean of all attention maps extracted by Eq.3, with the method DeiT and AAL as column (b) and (c), respectively.

 # D   Comparison for Learning Curve

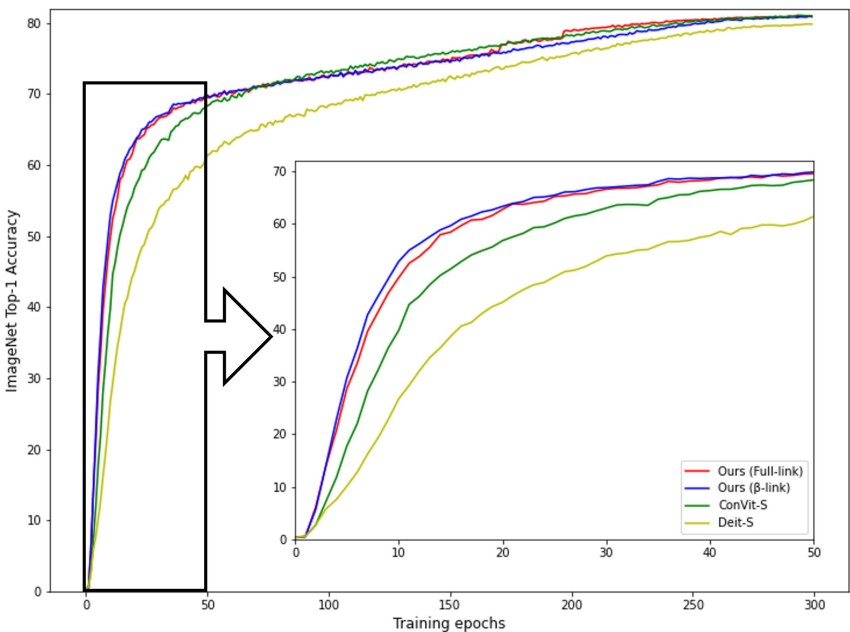

Figure 4: Learning curves in different methods during ImageNet training.

We emphasis the effect of our regularization method at early epochs with Figure 4. Our method
shows larger performance gap to other models at the beginning stage of learning. Also, the $\beta$-link
which showed relevance between ViT and CNN in ImageNet stably boosts training with having prior
connecting information compared to full-link setting.

 # E    Epoch-wise Change of Attention Links

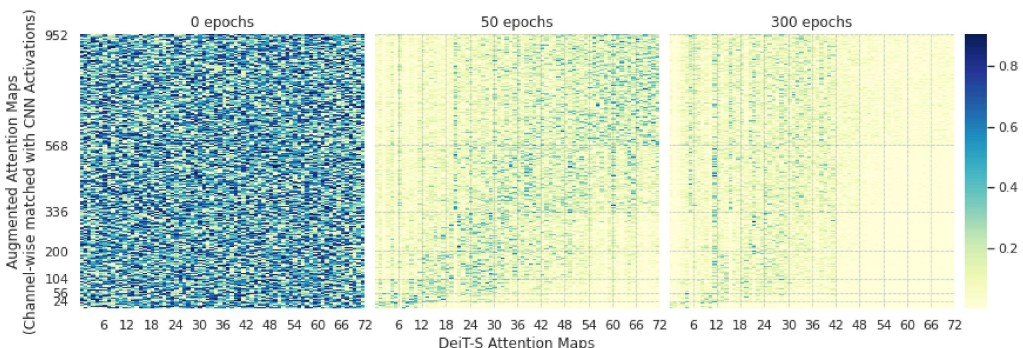

Figure 5: Variation of attention link during ImageNet training.

Figure 5 shows the transformation of adaptive links during training on ImageNet. At the end of the training, we observed that high-level heads are disconnected from augmented attention maps, meaning those are no more regularized by activations. This indicates that high-level attention heads escape locality and achieve long-range dependency which cannot be acquired by CNNs. From the analysis, we configured β-link setting to prevent high-level heads from over-regularization. Table 2 also proves the effectiveness of selective links.

 **F   Algorithm for Selective Link Extraction**

---

**Algorithm 1** Selective Link Extraction

---

**Input**: Trained adaptive link $w_c^{(m,n)} \in W$ where $c \in \{1, ..., C\}, m \in \{1, ..., M\}, n \in \{1, ..., N\}$
**Output**: Selective link $\Omega$
**procedure** LINK PRUNING($W$)
   $\Omega \leftarrow \{\}$
   **for** c = 1, ..., C **do**
      **for** m = 1, ..., M **do**
         **for** n = 1, ..., N **do**
            $w \leftarrow w_c^{(m,n)}$
            $w \leftarrow \frac{w - min(w)}{max(w) - min(w)}$
            **if** $|w| > \theta_l$ **then**
               $\Omega \leftarrow w_c^{(m,n)}$
            **end if**
         **end for**
      **end for**
   **end for**
   **return** $\Omega$
**end procedure**

---

 where $\theta_l$ is a hyperparameter, and we set $\theta_l$ to 0.05 for both the $\alpha$-link and $\beta$-link.

 **G   Model Flops and Parameter Size**

|         | DeiT-S       | ConViT-S | AAL             |
|---------|--------------|----------|-----------------|
| Params  | 22.4M        | 27.8M    | 22.5M           |
| FLOPs   | 4.27G        | 5.35G    | 4.27G           |
| Runtime | 0.40         | 1.23     | 0.36            |
| Teacher | RegNetY_160  | -        | EfficientNet_B3 |
| Params  | 83.6 M       | -        | 12.2 M          |
| FLOPs   | 15.9G        | -        | 0.98 G          |

Table 11: For runtime, we timed each model processing a batch with batch size 128 at training phase. This includes the forward pass of teacher CNN for both DeiT-S and AAL. DeiT-S and AAL shares similar value in number of parameters and FLOPs due to minimal design of attention augmentation module.