# OpenReview forum: "Adaptive Attention Link-based Regularization for Vision Transformers"
_NeurIPS.cc/2022/Conference — NeurIPS 2022 Submitted_

### Official Review · Reviewer_DJ1i · 2022-07-02

**Rating:** 6
**Confidence:** 4
**Soundness:** 3 good
**Presentation:** 3 good
**Contribution:** 4 excellent

**Summary:**

This paper proposes a new knowledge distillation method to improve the training efficiency of ViT. More concretely, the weighted average of the attention maps in the student ViT model is regularized to approach the activation maps of the CNN teacher model, which is the so-called attention link. Based on the empirical observations, the paper further makes the link selective by only building relations between CNN activation maps and ViT attention maps at similar levels and excluding the attention maps at the high levels. Empirically, the proposed method is compared with DeiT, ConViT and other knowledge distillation methods on ImageNet or CIFAR10. In the setting of small-sized data, the attention link is shown to improve the test accuracy with shorter training time and less computation cost.

**Questions:**

1. In Natural Language Processing, knowledge distillation has also been widely adopted in transformer-based models to enhance training efficiency. In particular, MiniLM (Wang et al., 2020) and TinyBert (Jiao et al., 2019) also focus on the regularization of attention maps. I think it’s worth including this line of literature in Related Work.
2. In Table 1, each CNN block level is associated with different numbers of ViT layer level. And beside the ViT layers at high level, the way how others are assigned to CNN block levels is also different between the alpha-link and the beta-link. I would appreciate it if there is any ablation study or verbal explanation on such a “non-intuitive” design.
3. In Table 2, could you please explain what the two numbers in each cell stand for? (as it’s not clearly written in the caption or the main text.)
4. In Figure 2, if my guess about the x-axis corresponding to the ViT layers is correct, I’m wondering how the weights of different heads within each layer are aggregated to produce the plot. Besides, as it’s hard to tell that the block diagonal is generally darker than the side in Figure 2(b) and (c), would you mind elaborating more on how to reach the conclusion that “the ViT attention maps are highly related to the CNN activation maps located at a similar level”?
5. I would appreciate it if the authors could explain more about the argument in line 239 - “the inductive bias is hard to be trained without a large dataset”.
6. As stated in Question 1, there are other knowledge distillations methods. For example, they may use a transformer as the teacher model instead of a CNN and also utilize the attention map regularization. I’m wondering whether it would be better to add the empirical comparison with these methods.

**Limitations:**

Please refer to the Weakness and Questions section for details.

**Strengths And Weaknesses:**

Strengths:
1. The writing of Sections 1-3 is clear and easy to follow.
2. The idea of regularizing the global token’s attention maps with the CNN activation maps is novel.
3. In the setting of small-sized data, the improvements in accuracy and training time are impressive.

Weaknesses:
(Please refer to the Questions section for more details.)
1. Some of the tables and figures lack elaboration in the caption.
2. More empirical evidence is needed to support the superiority of the attention link.

---

> ### Author Response · Authors · 2022-08-02
> **More empirical evidence to support the superiority of the attention link**
>
> We appreciate the reviewer found our work easy to follow, the idea is novel, with impressive results.
> We answered the reviewer's comments as follows.
>
> W1. Some of the tables and figures lack elaboration in the caption.
>
> We add the captions to tables and figures for readability, and all the boldfaced captions are entirely removed.
> If we have a chance to submit the camera-ready version, we are planning to hire professional proofreaders to improve the writing quality further.
>
> W2. More empirical evidence is needed to support the superiority of the attention link.
>
> As the empirical evidence to show the superiority of the proposed attention link, we perform three additional experiments.
> First, we empirically validate the effectiveness of the selective links including $\alpha$-link and $\beta$-link in the various scenarios with the comparison of the full link. $\alpha$-link showed 1.5% better Top-1 accuracy than $\beta$-link on ImageNet 5%. From the Figure 4 of supplementary materials, When the entire ImageNet dataset is used for training, $\beta$-link shows faster convergence than Full-link while their final accuracy becomes similar to each other. This result is reasonable because the connection to the high-level heads in Full-link can be disconnected through the update. These empirical results can verify the hypothesis since $\alpha$-link and $\beta$-link are based on the two hypotheses.
>
> Second, we additionally evaluate our framework through Weakly Supervised Object Localization (WSOL), which is frequently used to show space awareness~[1, 2]. WSOL trains the network model to classify the input image and evaluates the localization of target objects. We determine the position of the target objects by averaging the entire attention maps of Eq.3. We measure the localization performance by using the Intersection of Union (IoU) in CUB-200-2011, and the results are given in Table 10 in supplementary materials. We refer [2] for the evaluation method of WSOL. Compared to DeiT, although the proposed algorithm shows only 1% performance improvement in Table 9 in supplementary materials, its localization accuracy is 53% higher than that of DeiT with IoU threshold 0.5 in Table 10 in supplementary materials, which validates the superiority of the attention link for space awareness.
>
> Third, in Figure 4 of supplementary material, we compare the learning curve of our approach with those of DeiT and ConViT. We can see that our knowledge distillation achieves 70% accuracy at about 50 epochs while the DeiT needs 120 epochs to reach the same accuracy. This result validates the rapid convergence of our approach, which comes from the correlation of the attention links in our hypothesis.
>
> Finally, we enumerate the attention weight changes according to the training epochs. As shown in Figure 5 of supplementary material, the initial attention weight spread out randomly goes to become correlated by the similar depth levels of ViT and CNN models. Furthermore, after 300 epochs of training, the attention weights connecting the CNN models with the heads of last depth levels in ViT models become decreased, which represents the reduced influence of CNN activation maps on the high-level ViT attention maps. We added the visualization and the discussions in the revised manuscript and the supplementary material.
>
> [1] "Learning Deep Features for Discriminative Localization", Zhou et al. [2] "Evaluating Weakly Supervised Object Localization Methods Right", Choe et al.
>
>
> Q1. In Natural Language Processing, knowledge distillation has also been widely adopted in transformer-based models to enhance training efficiency. In particular, MiniLM (Wang et al., 2020) and TinyBert (Jiao et al., 2019) also focus on the regularization of attention maps. I think it’s worth including this line of literature in Related Work.
>
> We appreciate the reviewer's recommendation, and we add the previous studies to the related work section accordingly on lines 100-102. As the reviewer commented, the previous studies focus on the employment of knowledge distillation in natural language processing, so the target objectives are different from our proposed model.

---

> ### Author Response · Authors · 2022-08-02
> **Explanation on selective-link design**
>
> Q2. In Table 1, each CNN block level is associated with different numbers of ViT layer levels. Besides the ViT layers at a high level, the way how others are assigned to CNN block levels is also different between the alpha-link and the beta-link. I would appreciate it if there is any ablation study or verbal explanation on such a “non-intuitive” design.
>
> The weights of a full link are initialized by random values, so the training needs the initial computations to align the noisy weights letting the augmented attention maps be similar to the CNN activation map.
> In contrast, when we utilize the selective attention link, the noisy links can be ignored at the initial training phase, which results in reduced computation and stable training.
> This leads to the improved performance of the selective attention link, which also validates the analyzed correlation between the CNN activation map and the ViT attention map.
>
> In addition, we write down the pseudo-code for the selective attention link in the supplementary material.
>
>
> Q3. In Table 2, could you please explain what the two numbers in each cell stand for? (as it’s not clearly written in the caption or the main text.)
>
> To improve the readability, we entirely reorganize the tables and add captions to describe the values.
>
>
> Q4. In Figure 2, if my guess about the x-axis corresponding to the ViT layers is correct, I’m wondering how the weights of different heads within each layer are aggregated to produce the plot. Besides, as it’s hard to tell that the block diagonal is generally darker than the side in Figures 2(b) and (c), would you mind elaborating more on how to reach the conclusion that “the ViT attention maps are highly related to the CNN activation maps located at a similar level”?
>
> Yes. In figure 2, the x-axis corresponds to the ViT layers. To analyze the relations of two different architectures across layer depth, we head-wise averaged the absolute scale of link weights. We agree that those heatmaps are not strictly diagonal but show correlation along relative layer depth. This accords with previous work [*], which used cross-model Centered Kernel Alignment to check representation similarity between ViT and ResNet across layers. We add the captions to describe the plots.
>
> [*] Maithra Raghu, Thomas Unterthiner, Simon Kornblith, Chiyuan Zhang, and Alexey Dosovitskiy. Do Vision transformers see like convolutional neural networks? ArXiv, abs/2108.08810, 2021.691
>
>
> Q5. I would appreciate it if the authors could explain more about the argument in line 239 - “the inductive bias is hard to be trained without a large dataset”.
>
> From the sentence, we tried to emphasize the need for a large dataset for Vision Transformer without the inductive bias. A similar statement was mentioned in L21-L24. To clear out the misunderstanding, we revised the argument as 'semantic information that ignores the inductive bias is hard to be trained without a large dataset'.
>
>
> Q6. As stated in Question 1, there are other knowledge distillations methods. For example, they may use a transformer as the teacher model instead of a CNN and also utilize the attention map regularization. I’m wondering whether it would be better to add the empirical comparison with these methods.
>
> We agree that it is a probable approach. However, we believe our source of performance gain is due to transferring CNN's inductive bias with attention. Also, while one of our contributions is to reduce ViT's lengthy training time, using transformers as a teacher model would require more FLOPs and longer runtime.

---

> ### Comment · Reviewer_DJ1i · 2022-08-04
> **Thank you for the response.**
>
> I appreciate the authors' detailed responses. My doubts / confusions have been cleared up.

---

> > ### Author Response · Authors · 2022-08-06
> > **Dear reviewer DJ1i**
> >
> > We appreciate your feedback, and we are happy that our response was helpful to resolve your concerns and questions.
> >
> > Best regards,
> > Authors

---

### Official Review · Reviewer_StQc · 2022-07-07

**Rating:** 6
**Confidence:** 4
**Soundness:** 3 good
**Presentation:** 2 fair
**Contribution:** 3 good

**Summary:**

This paper introduces a new method to increase data efficiency for Vision Transformers (VTs). The idea of the method is that Adaptive Attention Links (AAL) between the convolutional filters of a CNN and between the attention heads of the VT are trainable parameters which connect these feature maps between the teacher CNN and the student VT, with only a small parameter increase. This way, the locality inductive bias can be learned from the teacher model, decreasing training time. The paper confirms previous results that higher layer attention maps in a VT contain more high-level semantic information than a CNN, but that other layers roughly correspond to the same type of hierarchical information. Extensive experiments show that the method improves on previous distillation approaches between a CNN and a VT.

**Questions:**

1. How was lambda = 0.99 and 0.98 chosen? (L288)
2. In lines 37-43, do you mean source dataset instead of target dataset?
3. L175-176, did you find an improvement when using channel-wise attention maps compared to some aggregate version of attention maps?
4. What is your interpretation of the augmented attention links, what do they represent? Is it mainly about their increased flexibility of representing the CNN maps?
5. Are the X/Y results in the tables always teacher/student?
6. What does attention maps “at a similar level” mean? At a similar layer depth?
7. L266, when you say remarkable results, which results are you referring to then? If you say a strong word such as remarkable it should be clear what you mean, apart from that the result should be strong too, of course.
8. L327-332: do you compare self-supervised approaches to your classically supervised approach here? Or do you make some kind of SS version of your approach? This is not clear


**Limitations:**

The choice of the hyper parameter lambda could be more accounted for (did it include tuning, etc). The authors do not state any potential limitations of their work, which seems a little worrying, perhaps, in terms of transparency. There are always limitations. However, if this can be added the work is sound.

**Strengths And Weaknesses:**

S1: The idea of the paper is novel, to the best of my knowledge.
S2: The idea is clear and conceptually appealing.
S3: The paper presents extensive experimentation to support its conclusions (a number of different datasets and variants of models).
S4: The paper promotes better data efficiency for VTs, which is a well-known issue.
S5: The paper presents ablations and one example of repeated runs which show stable results (this seems reasonable in order to save computation).
S6: the method is flexible with regards to the VT student architecture, any model containing hierarchical attention maps can be used, as well as wrt the teacher model: any CNN containing hierarchical feature maps can be used. The down-stream task of the teacher model does not matter.

W1 The main weakness of the paper, in my view, is its clarity. This can be straightforwardly fixed by doing a more thorough language check, for example by using Grammarly or similar. Many sentences are hard to parse, eg lines 320-321, 234-239. Furthermore, the captions to tables and figures should be more informative which would help the reading a lot, right now they are not possible to get the gist from without finding their reference in the text. Also, captions should not be boldfaced, but that is a detail of course.

W2, W3: Other weaknesses are lacking mention of how the hyper parameters lambda were chosen, and a lacking discussion of the potential weaknesses of the method— this should be easily amendable.

W4: No code is released.

---

> ### Author Response · Authors · 2022-08-02
> **Choosing hyperparameters lambda**
>
> We appreciate that the reviewer said our idea is novel, clear, and conceptually appealing. In addition, the reviewer commented that our extensive experiment and ablation studies support our conclusion well. As the reviewer mentioned, our approach is flexible with the student ViT models and the teacher CNN models.
>
>
> W1. The main weakness of the paper, in my view, is its clarity.
>
> We appreciate the recommendation, and we revise the commented sentences accordingly. In addition, we add the captions to tables and figures for readability, and all the boldfaced captions are entirely removed. If we have a chance to submit the camera-ready version, we are planning to hire professional proofreaders to improve the writing quality further.
>
>
> W2. Another weakness are lacking of mention of how the hyperparameters lambda were chosen
>
> At every experiment, we set $\lambda$ to $2000$, and we are sorry for the missing initialization value of $\lambda$.
> To justify the value choice, we perform the ablation studies by changing the value of $\lambda$ by $1500$, $2000$, and $2500$.
> In ImageNet 10%, we obtain top-1 accuracy of 56.5, 64.7, and 64.6 respectively for lambda=1500, 2000, 2500, and we acquire the best performance when $\lambda$ is set to $2000$.
> Since $\lambda$ controls the scale of the regularization loss term, its value highly correlates with the overfitting and the underfitting of the target model. As a result, the value of $\lambda$ should be set to avoid both the overfitting and the underfitting issues, and we empirically found that the value of $\lambda$ is a proper choice. Due to the lack of time, we fail to show various results, but we will supplement the ablation study with the additional values of  in the final version.
>
> W3. and a lacking discussion of the potential weaknesses of the method
>
> Even with its training efficiency, the loss term of $L_{att}$ could feed the inductive bias of CNN models into the ViT models, which results in unnecessary regularization when sufficient training data is given.
> We ensure that our approach shows slightly less performance than ConViT because of this issue when the full ImageNet dataset is considered.
> Even though we relieve the issue by employing the decay rate of $\lambda$, additional adaptation is required to effectively control the power of inductive bias from the CNN models.
> We are planning to solve the problem in our future work.
>
> The current framework can only handle one teacher model that is limited by one of the CNN models.
> In contrast, when we can integrate multiple teacher models trained by various datasets, the performance generalization would be further improved.
> In addition, the current model cannot consider the ViT-based teacher model, which limits the variety of the teacher model.
> Due to the simplicity of this work, we guess that employing the multiple teacher models is a reasonable approach, so we will extend this algorithm for the multiple teacher models even including the transformer-based models.
> If we have a chance to submit the camera-ready version, we will add these limitations to the paper.
>
> W4: No code is released.
>
> We release our code willingly if the paper is accepted.
>
> Q1. How was lambda = 0.99 and 0.98 chosen? (L288)
>
> As shown in Figure 5 of supplementary material, the ViT attention maps seem similar to the CNN activation maps at the related depth levels, while the relation becomes weakened as training goes on.
> Accordingly, we designed our approach to reduce the power of the attention-based knowledge distillation loss terms by employing the decay constant of $\lambda$.
> In addition, we perform the ablation study where the variant is built by fixing the decay constant with $0.99$.
> In ImageNet 50\%, the variant with the fixed decay constant shows the performance of $77.8\%$ and $93.8\%$ for top-1 and top-5 accuracy, respectively, which are less than $78.9\%$ and $94.5\%$ of the proposed algorithm.
> This result shows that we can tune the decay constant to improve the efficiency of our approach, which would be analyzed in our future work.
>
>
>
> Q2. In lines 37-43, do you mean source dataset instead of target dataset?
>
> As the reviewer commented, the target dataset is a source dataset to train the final model.
> However, we found that the sentence is ambiguous to deliver our meaning, so we revised the entire sentence as follows:
> However, the previous studies have the remaining limitations where the training datasets must be equivalent for both the student and teacher models.

---

> ### Author Response · Authors · 2022-08-02
> **Interpretation of the augmented attention links**
>
> Q4. What is your interpretation of the augmented attention links, what do they represent?
>
> For the detailed interpretation of the augmented attention links, we perform three additional experiments.
> First, we additionally evaluate our framework through Weakly Supervised Object Localization (WSOL), which is frequently used to show space awareness~[1, 2].
> WSOL trains the network model to classify the input image and evaluates the localization of target objects.
> We determine the position of the target objects by averaging the entire attention maps of Eq.3.
> We measure the localization performance by using the Intersection of Union (IoU) in CUB-200-2011 and Oxford Flower 102 datasets, and the results are given in Table 9 in supplementary materials. We refer [2] for the evaluation method of WSOL.
> Compared to DeiT, although the proposed algorithm shows only 1\% performance improvement, its localization accuracy is 53% higher than that of DeiT with IoU threshold 0.5, which validates the superiority of the attention link for spatial awareness.
>
> Second, in Figure 4 in supplementary materials, we compare the learning curve of our approach with those of DeiT and ConViT.
> We can see that our knowledge distillation achieves 70\% accuracy at about 50 epochs while the two comparisons need more epochs for the same accuracy.
>
> Finally, we enumerate the attention weight changes according to the training epochs.
> As shown in Figure 5 in supplementary materials, the initial attention weight spread out randomly goes to become correlated by the similar depth levels of ViT and CNN models.
> Furthermore, after 300 epochs, the attention weights connecting the CNN models with the heads of last depth levels in ViT models become decreased, which represents the reduced influence of CNN activation maps on the high-level ViT attention maps.
> We added the visualization and the discussions in the revised manuscript and the supplementary material.
>
> [1] "Learning Deep Features for Discriminative Localization", Zhou et al. [2] "Evaluating Weakly Supervised Object Localization Methods Right", Choe et al.
>
> Q5. Are the X/Y results in the tables always teacher/student?
>
> A5.
> In Table 2, X/Y indicate Top-1/Top-5 test accuracy, respectively.
> In the column 'Gain' of Table 4, X and Y represent the performance gap between the previous study and ours.
> To clear out the misunderstanding, we entirely revise the structure of the tables and add the captions for the description.
>
>
> Q6. What does attention maps “at a similar level” mean? At a similar layer depth?
>
> A6. Yes, "attention maps at a similar level” means the attention maps selected at similar layer depth of the respective architecture.
>
>
> Q7. L266, when you say remarkable results, which results are you referring to then?
>
> A7.
> We agree that the word 'remarkable' should be used more carefully.
> We refer to the remarkable results for the performance in the low-data regime, which outperforms DeiT and ConViT with large performance gaps. However, we think that the order of the table may cause your misunderstanding, so we reorganized the experimental sections and removed the strong words.
>
>
> Q8. L327-332: do you compare self-supervised approaches to your classically supervised approach here? Or do you make some kind of SS version of your approach? This is not clear
>
> A8. We compared self-supervised approaches to our classically supervised approach. We have yet to consider a self-supervised version of our approach, but we believe it is possible for our future work.

---

> ### Author Response · Authors · 2022-08-06
> **Dear reviewer StQc,**
>
> As the end of the open discussion gets closer, we would like to gently remind you to read our rebuttal, which should hopefully answer all of your concerns and comments on the reviews.
>
> If you have any comments or any questions, we would be happy to address them.
>
> Sincerely,
>
> Authors

---

> > ### Comment · Reviewer_StQc · 2022-08-09
> > **Thank you**
> >
> > Thank you to the authors for your detailed replies to my questions. Having cleared my confusions, I recommend publication for the revised version of the article.

---

> > ### Author Response · Authors · 2022-08-09
> > **Dear reviewer StQc**
> >
> > The authors appreciate your thorough feedback and recommendation for the revised version of the article. We are happy that our answer to your questions have cleared the confusions.
> >
> > Best regards, Authors

---

### Official Review · Reviewer_Fq3N · 2022-07-11

**Rating:** 6
**Confidence:** 3
**Soundness:** 3 good
**Presentation:** 3 good
**Contribution:** 3 good

**Summary:**

The paper introduces a novel regularisation method to train ViT. Specifically, they rely on the activation features from a convolutional neural network to transfer the knowledge and augment the training of the vision transformer. Furthermore, they discuss how to select which links to keep during training in order to make the learning more efficient. Finally, they ablate and evaluate the approach on image datasets such as ImageNet and CIFAR.

**Questions:**

My concerns are listed in the weaknesses section. I would expect authors to discuss:

- Include additional metrics in the table for a fair comparison with the baselines. What is the effect of the cost of running the CNN?

- Have the authors considered other task beyond classification?

- Have the authors considered using other datasets?

- Have the authors considered including more qualitative analysis of the learned model compared to the baseline?

**Limitations:**

Although in the form authors claim that Section 6 discussed limitations, I do think it would be necessary to extend this discussion for the paper to be accepted. I see Section 6 as a conclusion but no specific mention to the limitation is done.

**Strengths And Weaknesses:**

**Strengths**:

- S1. The approach is simple and effective. Authors join the advantages of CNNs and ViT with their augmentation procedure in order to make learning more efficient. I really like the simplicity of the approach, which also would make it more likely to be used by the community.

- S2. Analysing the most important links is also one extra step in terms of efficiency. The paper does a good job introducing the overall intuition and the general method and then going into link selection.

- S3. The results show how the approach can be specially useful for low data regimes.

- S4. Most of previous works using transformers specially rely on very large datasets while this paper tackles the problem of training in a low data regime.

- S5. Authors provide sufficient experimental evidence to answer the main questions in the paper. The ablation study is complete and useful for the reader.

**Weaknesses**
- W1. In my opinion the paper avoid the issue of the additional overhead of running the CNN alongside the transformer training. I think the comparison table should take that into account and report FLOPS or runtime for a complete picture.

- W2. The methodology has only been used for classification. However, it seems like other task which require for awarness of the space would be a better fit for such a paper. I wonder if authors have considered evaluating on additional task.

- W3. Authors evaluate on CIFAR and ImageNet, which are very saturated dataset at this point. Although I think the point of the paper is valid, I would appreciate additional evaluations.

- W4. I am missing some qualitative comparison on *why* the techniques improve data efficiency. What is different in the learning curves? How do the attention weights change?

- W5. Have the authors played a bit with the CNN architecture? I wonder if they could safe computation by running a very basic model.

---

> ### Author Response · Authors · 2022-08-02
> **Comparison table with additional task**
>
> We appreciate your comments that our approach is simple but effective. As the reviewer mentioned, we are expecting that the community can employ our approach to improve the training stability and efficiency of the ViT-based models. Furthermore, the positive responses to our analysis and results have encouraged us to perform additional experiments with passion. As the reviewer commented, we focus on the problem of ViT-based models in a low data regime, and we show reasonable results to solve the issue by using the attention maps of CNN models. We will add the discussion about limitations in the camera-ready version.
>
> W1. I think the comparison table should take that into account and report FLOPS or runtime for a complete picture.
>
> |  | DeiT-S | ConViT-S | AAL |
> |---|---|---|---|
> | Params | 22.4M | 27.8M | 22.5M |
> | FLOPs | 4.27G | 5.35G | 4.27G |
> | Runtime | 0.40 | 1.23 | 0.36 |
> | Teacher | RegNetY_160 | - | EfficientNet_B3 |
> | Params | 83.6 M | - | 12.2 M |
> | FLOPs | 15.9G | - | 0.98 G |
>
> According to the reviewer's comment, we observed the training runtime and inference FLOPS.
> We measure the runtime by measuring the time for processing a batch with batch size 128 at the training phase.
> DeiT-S and AAL shares similar value in the number of parameters and FLOPs due to the minimal design of the attention augmentation module.
> As shown in table above, our framework shows the fastest runtime among the three comparisons including DeiT and ConViT, which validates the training efficiency of our framework.
> Furthermore, our approach needs 20.2\% fewer inference FLOPS than ConViT even though they have similar performance.
>
> W2. I wonder if authors have considered evaluating on additional task.
>
> According to the reviewer's comment, we additionally evaluated our framework through Weakly Supervised Object Localization (WSOL), which is frequently used to show space awareness~[1]. WSOL trains the network model to classify the input image and evaluates the localization of target objects.
> We determine the position of the target objects by averaging the entire attention maps of Eq.3.
> We measure the localization performance by using the Intersection of Union (IoU) in CUB-200-2011. The results is shown in Table 10 of supplementary materials.
> We refer [2] for the evaluation method of WSOL.
> Compared to DeiT, although the proposed algorithm shows only 1\% performance improvement, its localization accuracy is 53% higher than that of DeiT with IoU threshold 0.5, which validates the space awareness of our knowledge distillation scheme.
>
> [1] "Learning Deep Features for Discriminative Localization", Zhou et al.
> [2] "Evaluating Weakly Supervised Object Localization Methods Right", Choe et al.

---

> > ### Author Response · Authors · 2022-08-02
> > **Additional evaluations**
> >
> > W3. I would appreciate additional evaluations.
> >
> > We perform the additional experiments by using CUB-200-2011 and Oxford Flowers 102 datasets, which are challenging due to their enormous categories and the lack of class-wise training data.
> > As shown in Table 9 in supplementary material, the proposed framework shows much better performance than the previous studies, which validates its generality across various scenarios.
> > Especially, our approach shows an enlarged performance gap of 92.7\% when the ViT model is randomly initialized, which shows its robustness in the absence of a well-trained model.
> >
> > W4. I am missing some qualitative comparisons on why the techniques improve data efficiency.
> >
> > As recommended by the reviewer, we qualitatively visualize the learning curve and the attention weights change to show the data efficiency.
> > In Figure 4 of supplementary material, we compare the learning curve of our approach with those of DeiT and ConViT. We can see that our knowledge distillation achieves 70\% accuracy at about 50 epochs while the two comparisons need at most 120 epochs for the same accuracy.
> > In addition, we enumerate the attention weight changes according to the training epochs.
> > As shown in Figure 5 of supplementary material, the initial attention weight spread out randomly goes to become correlated by the similar depth levels of ViT and CNN models.
> > Furthermore, after 300 epochs, the attention weights connecting the CNN models with the heads of last depth levels in ViT models become decreased, which represents the reduced influence of CNN activation maps on the high-level ViT attention maps.
> > We added the visualization and the discussions in the revised manuscript and the supplementary material.
> >
> > W5. Have the authors played a bit with the CNN architecture?
> >
> > We use EfficientNet-b3 and ResNet34 as our teacher models, which are much smaller models than RegNetY-160 which is a teacher model of DeiT.
> > Even though EfficientNet-b3 needs only 14.63\% parameters compared with RegNetY-160, our model achieves better performance than DeiT, which validates its impressive training efficiency.
> > However, to resolve the reviewer's concern, we perform the additional experiments where the teacher model of our approach is replaced by ResNet-18.
> > For ImageNet 5\%, the variant results in 55.0\% and 78.9\% for top-1 and top-5 accuracy, respectively.
> > Thus, even though ResNet-18 needs only 13.98\% parameters compared with RegNetY-160, our approach still suppresses the performance of DeiT (34.8\%\&57.8\%) by showing the performance gaps of 58\% and 37\%.
> >
> > Limitations: Lacking discussion of the potential limitations;
> >
> > Even with its training efficiency, the loss term of $L_{att}$ could feed the inductive bias of CNN models into the ViT models, which results in unnecessary regularization when sufficient training data is given.
> > We ensure that our approach shows slightly less performance than ConViT because of this issue when the full ImageNet dataset is considered.
> > Even though we relieve the issue by employing the decay rate of $\lambda$, additional adaptation is required to effectively control the power of inductive bias from the CNN models.
> > We are planning to solve the problem in our future work.
> >
> > The current framework can only handle one teacher model that is limited by one of the CNN models.
> > In contrast, when we can integrate multiple teacher models trained by various datasets, the performance generalization would be further improved.
> > In addition, the current model cannot consider the ViT-based teacher model, which limits the variety of the teacher model.
> > Due to the simplicity of this work, we guess that employing the multiple teacher models is a reasonable approach, so we will extend this algorithm for the multiple teacher models even including the transformer-based models.
> > If we have a chance to submit the camera-ready version, we will add these limitations to the paper.

---

> > > ### Comment · Reviewer_Fq3N · 2022-08-09
> > > **Thanks!**
> > >
> > > Dear Authors,
> > >
> > > After having read the rebuttal in detail, my concerns have been addressed and I recommend the acceptance of the paper.
> > >
> > > Best wishes.

---

> > > > ### Author Response · Authors · 2022-08-09
> > > > **Dear reviewer Fq3N**
> > > >
> > > > The authors appreciate the reviewer for their detailed review of our manuscript and positive feedback. We are happy that our response has addressed your concerns.
> > > >
> > > > Best regards,
> > > >
> > > > Authors

---

> ### Author Response · Authors · 2022-08-06
> **Dear reviewer Fq3N,**
>
> As the end of the open discussion gets closer, we would like to gently remind you to read our rebuttal, which should hopefully answer all of your concerns and comments on the reviews.
>
> If you have any comments or any questions, we would be happy to address them.
>
> Sincerely,
>
> Authors

---

> ### Comment · Area_Chair_XLKL · 2022-08-09
> **Rebuttal Response**
>
> Dear Reviewer,
>
> could you please indicate that you have considered the authors' rebuttal? (E.g. by replying to the rebuttal or at least by using the "Author Rebuttal Acknowledgement".)
>
> The [reviewer guidelines](https://nips.cc/Conferences/2022/ReviewerGuidelines) ask: "Even if the author response didn’t change your opinion about the paper, please acknowledge that you have read and considered it."
>
> Thank you!

---

### Official Review · Reviewer_xcGt · 2022-07-11

**Rating:** 3
**Confidence:** 4
**Soundness:** 2 fair
**Presentation:** 1 poor
**Contribution:** 2 fair

**Summary:**

This paper proposes a loss regularization method for training Vision Transformer (ViT) through matching the attention maps of ViT with the activation map of a pre-trained CNN. The authors aggregate ViT's attention maps from different layers into the number of CNN's activation maps and then use their L2 norm as the additional loss regularization of training ViT. It is shown that the proposed method can outperform the selected baseline on ImageNet when the number of training samples is small.

**Questions:**

See weaknesses above.

**Limitations:**

I didn't check but I don't think this work will lead to a potential negative societal impact

**Strengths And Weaknesses:**

Strengths
The idea of matching attention maps of ViT with the activation maps looks interesting, and the motivation behind this seems reasonable.

Weakness
1. How do the authors learn the w_c in Eq. 4, is it end-to-end trainable with Eq. 6 or learned separately? Besides, do the attention map (A_c) and activation map (B_c) in Eq. 5 have the same shape? If not, how do you address it?
2. The computation complexity of Eq. 4 is also a concern. Since w_c is posed for each pixel in A_(m,n), the complexity of w_c will grow exponentially as P or the number of layers in ViT grows. How will it perform when facing larger or deeper ViT models?
3. I have checked the manuscript, but I cannot find the initialization value of λ in Eq. 6. Please include it and justify the value choice as it is an important hyper-parameter.
4. The current description of the experimental setting and results, in my opinion, is difficult for the readers to understand. Please re-organize it - first describe the dataset and different experiment settings, and then explain the results one by one.
5. I don't think Figure 2 is sufficient to support the two hypotheses in 4.1. Besides, the selective attention link should be discussed in more detail, e.g. why it can achieve better results than the full link.
6. Please clarify whether the baseline model in Table 4 has been trained with a teacher model or not. If not, the comparison may not be fair. Besides, it seems the performance cannot outperform the ConViT with the full train set, and therefore the generality to large-scale data is in doubt.

Overall, although the idea of this paper is interesting, the methodology is not explained or designed properly, the descriptions of the current experimental settings and results are difficult to read, and also the achieved performance is not convincing. I don't think this paper can be accepted, and I tend to vote for rejection.

---

> ### Author Response · Authors · 2022-08-02
> **Attention link is end-to-end trainable. Activation maps are resized with bi-cubic interpolation. We set $\lambda$ to $2000$ at every experiment scenario.**
>
> We appreciate the reviewer's comment that our method is interesting and its motivation is reasonable. We respond to every comment with the additional experiments, and we revised the paper and added the supplementary material accordingly.
>
> Q1. Is $w_c$ in Eq.4 end-to-end trainable with Eq.6?
>
> A1. Yes, the adaptive attention link that is $w_c$ in Eq.4 is trained in the end-to-end scheme with Eq.6.
> Since $A_c^+$ is differentiable by $w_c$, $L_{att}$ of Eq.5 directly updates $w_c$ to reduce the L2 distance between the CNN activation map and the augmented attention map .
> Meanwhile, $L_{ce}$ also affects indirectly the update of $w_c$ due to its influence on the ViT attention map ($A$) in Eq.4.
>
> Q2. Do the attention map ($A_c$) and activation map ($B_c$) in Eq.5 have the same shape?
>
> A2. From Eq.3, the spatial size of ViT attention map ($A_c$) is defined by $(P\times P)$ where $P^2$ is the number of initial input patches.
> Thus, we resize each CNN activation map ($B_c$) by using bi-cubic interpolation to be $(P\times P)$.
> The resizing steps are described in lines 179-181, but we revise the sentence to mention the bi-cubic interpolation.
>
> Q3. The computation complexity of Eq.4 is also a concern. Since $w_c$ is posed for each pixel in $A_(m,n)$, the complexity of $w_c$ will grow exponentially.
>
> A3. The sub-indices $(m,n)$ of attention link $w_c$ and the attention maps $A_c$ are the indices for the self-attention head and level depth, respectively.
> Thus, the size of $w_c$ is determined by the channel depth size of the CNN model and the number of self-attention heads in ViT model, which is irrelevant to the spatial size of the attention map.
> We found that the missing definitions of $M$ and $N$ cause ambiguity, so we add the definitions below Eq.4.
> Furthermore, as we mentioned in line 298, $w_c$ needs only 0.068M additional parameters which are quite small compared to the parameter size of conventional ViT models.
>
> Q4. Please include the initialization value of $\lambda$ in Eq.6 and justify the value choice as it is an important hyper-parameter.
>
> A4. We set $\lambda$ to $2000$ at every experiment scenario, and we are sorry for the missing initialization value of $\lambda$.
> To justify the value choice, we perform the ablation studies by changing the value of $\lambda$ by $1500$, $2000$, and $2500$ for ImageNet 10\%.
> In ImageNet 10%, we obtain top-1 accuracy of 56.5, 64.7, and 64.6 respectively for lambda=1500, 2000, 2500, and we acquire the best performance when $\lambda$ is set to $2000$.
> Since $\lambda$ controls the scale of the regularization loss term, its value highly correlates with the overfitting and the underfitting of the target model.
> As a result, the value of $\lambda$ should be set to avoid both the overfitting and the underfitting issues, and we empirically found that the value of $2000$ is a proper choice.
> Due to the lack of time, we fail to show various results, but we will supplement the ablation study with the additional values of $\lambda$ in the final version.

---

> ### Author Response · Authors · 2022-08-02
> **Detailed interpretation of the augmented attention links**
>
> Q5. First describe the dataset and different experiment settings, and then explain the results one by one.
>
> A5. We appreciate the recommendation, and we entirely rewrote and reorganized the experiment section accordingly. The reviewer can refer to the red lines of the submitted revised paper.
>
> Q6. I don't think Figure 2 is sufficient to support the two hypotheses in 4.1.
>
> A6. For the detailed interpretation of the augmented attention links, we perform four additional experiments.
>
> First, we empirically validate the effectiveness of the selective links including $\alpha$-link and $\beta$-link in the various scenarios with the comparison of the full link. $\alpha$-link showed 1.5\% better Top-1 accuracy than $\beta$-link on ImageNet 5\%.
> From the Figure 4 of supplementary materials, When the entire ImageNet dataset is used for training, $\beta$-link shows faster convergence than Full-link while their final accuracy becomes similar to each other.
> This result is reasonable because the connection to the high-level heads in Full-link can be disconnected through the update.
> These empirical results can verify the hypothesis since $\alpha$-link and $\beta$-link are based on the two hypotheses.
>
> Second, we additionally evaluate our framework through Weakly Supervised Object Localization (WSOL), which is frequently used to show space awareness~[1, 2].
> WSOL trains the network model to classify the input image and evaluates the localization of target objects.
> We determine the position of the target objects by averaging the entire attention maps of Eq.3.
> We measure the localization performance by using the Intersection of Union (IoU) in CUB-200-2011, and the results are given in Table 10 in supplementary materials.
> We refer [2] for the evaluation method of WSOL.
> Compared to DeiT, although the proposed algorithm shows only 1\% performance improvement in Table 9 in supplementary materials, its localization accuracy is 53\% higher than that of DeiT with IoU threshold 0.5 in Table 10 in supplementary materials, which validates the superiority of the attention link for space awareness.
>
> Third, in Figure 4 of supplementary material, we compare the learning curve of our approach with those of DeiT and ConViT.
> We can see that our knowledge distillation achieves 70\% accuracy at about 50 epochs while the DeiT needs 120 epochs to reach the same accuracy. This result validates the rapid convergence of our approach, which comes from the correlation of the attention links in our hypothesis.
>
> Finally, we enumerate the attention weight changes according to the training epochs.
> As shown in Figure 5 of supplementary material, the initial attention weight spread out randomly goes to become correlated by the similar depth levels of ViT and CNN models.
> Furthermore, after 300 epochs of training, the attention weights connecting the CNN models with the heads of last depth levels in ViT models become decreased, which represents the reduced influence of CNN activation maps on the high-level ViT attention maps.
> We added the visualization and the discussions in the revised manuscript and the supplementary material.
>
> [1] "Learning Deep Features for Discriminative Localization", Zhou et al.
> [2] "Evaluating Weakly Supervised Object Localization Methods Right", Choe et al.
>
> Q7. The selective attention link should be discussed in more detail, e.g. why it can achieve better results than the full link.
>
> A7. The weights of the full link are initialized by random values, so the training needs the initial computations to align the noisy weights letting the augmented attention maps be similar to the CNN activation map.
> In contrast, when we utilize the selective attention link, the noisy links can be ignored at the initial training phase, which results in reduced computation and stable training.
> This leads to the improved performance of the selective attention link, which also validates the analyzed correlation between the CNN activation map and the ViT attention map.
>
> In addition, we write down the pseudo-code for searching the selective attention link in supplementary material.

---

> ### Author Response · Authors · 2022-08-02
> **Baseline models with teacher models**
>
> Q8. Clarify whether the baseline model in Table 4 has been trained with a teacher model or not. If not, the comparison may not be fair.
>
> A8. Among the compared algorithms, DeiT is trained with a teacher model of RegNetY-16GF, which is a CNN model showing better performance than our teacher models such as EfficientNet-b3 and ResNet-34.
> Even though the base model of ConViT does not use the teacher model, their supplementary material reports that the hard knowledge distillation for ConViT shows an ignorable improvement in performance compared to accuracy gain in DeiT.
>
> In addition, we show comparision table for baseline models.
>
> |  | DeiT-S | ConViT-S | AAL |
> |---|---|---|---|
> | Params | 22.4M | 27.8M | 22.5M |
> | FLOPs | 4.27G | 5.35G | 4.27G |
> | Runtime | 0.40 | 1.23 | 0.36 |
> | Teacher | RegNetY_160 | - | EfficientNet_B3 |
> | Params | 83.6 M | - | 12.2 M |
> | FLOPs | 15.9G | - | 0.98 G |
>
> We measure the runtime by measuring the time for processing a batch with batch size 128 at the training phase.
> DeiT-S and AAL shares similar value in the number of parameters and FLOPs due to the minimal design of the attention augmentation module.
> As shown in table above, our framework shows the fastest runtime at training phase which includes the inference of teacher CNN among the three comparisons including DeiT and ConViT. This additionally validates the training efficiency of our framework.
> Furthermore, our approach needs 20.2\% fewer inference FLOPS than ConViT even though they have similar performance.
>
> Q9. It seems the performance cannot outperform the ConViT with the full train set, and therefore the generality to large-scale data is in doubt.
>
> A9.
> As we mentioned in the introduction section, our main objective is to improve the data efficiency and training load of ViT models.
> Accordingly, we successfully show the impressive gain in performance when insufficient data is given for the training as shown in the first rows in Table 2.
> We show the experimental results using the large-scale data to verify that the proposed algorithm is not harmful even for the large-scale data, which is important for its generality to the various scenarios.
> Interestingly, our model shows similar performance to ConViT even though its model size is 22.3\% less than that of ConViT, which also verifies its training efficiency.

---

> ### Author Response · Authors · 2022-08-06
> **Dear reviewr xcGt**
>
> As the end of the open discussion gets closer, we would like to gently remind you to read our rebuttal, which should hopefully answer all of your concerns and comments on the reviews.
>
> If you have any comments or any questions, we would be happy to address them.
>
> Sincerely,
>
> Authors

---

> ### Comment · Reviewer_xcGt · 2022-08-09
> **Thanks for the detailed response**
>
> I thank the authors for the detailed response, which have cleared parts of my initial concerns. However, I am still concerned about whether the revised manuscript can be improved significantly and satisfyingly in such a short time, with so many modifications to make.

---

> > ### Author Response · Authors · 2022-08-09
> > **Dear reviewer xcGt**
> >
> > The authors appreciate your feedback and are happy that your initial concerns have been resolved.
> > We updated the revised version of the paper with supplementary materials which are currently accessible on this page.
> > The authors would be thankful to the reviewer if the reviewer could check the revised version of the article.
> > As the reviewer commented, during the short rebuttal time, we had no choice other than to focus on the revision of experimental parts in the paper.
> > However, we are continually working on polishing the paper with a thorough language check, and we are sure that we can finalize the revised version according to the feedback from every reviewer.
> > If we have a chance to submit the camera-ready version, we are planning to hire professional proofreaders to improve the writing quality further.
> >
> > Best regards,
> >
> > Authors

---

### Meta-Review · Area_Chair_XLKL · 2022-08-25

**Recommendation:** Reject
**Confidence:** Less certain

**Metareview:**

Four reviewers provided detailed feedback on this paper. The authors responded to the reviews and I appreciate the authors' comments and clarifications, specifically that each question/comment is addressed in detail. Additional experiments were also performed. The authors also uploaded a revised version of the paper.

After the two discussion periods, one of the four reviewers suggest to reject the paper while three reviewers rate the paper as "weak accept", so no reviewer strongly advocates for acceptance. I considered the reviewers' and authors' comments and also tried to assess the paper directly. I believe that the paper should not be accepted to NeurIPS in its current form.

Weaknesses include:
* Readability: While at least one reviewer describes part of the paper as "clear and easy to follow" one other reviewer mentions clarity as the main weakness and one other reviewer also comments in this direction. I personally found the paper hard to read as well (even after the improvements made in the revision), and I found some of the claims to be fairly generic and partially not well supported. E.g. "resolve the issues of overfitting and lengthy training time of ViT", or "The proposed scheme preserves the original architecture of ViT, which results in its general employment regardless of the architecture of ViT."
* Experimental Results: Several questions have been raised regarding the experimental results (e.g. influence of the attention link, choice of hyperparameters). These have been addressed in the discussion, but it seems to me that they were at best partially resolved.
* Relation to distillation: The results in the low-data regime rely on learning from a teacher model. This relation to distillation is recognized but somewhat under-explored. This could be a confounding factor in the analysis of the approach. For example, in one response, the authors argue that "However, we believe our source of performance gain is due to transferring CNN's inductive bias with attention". It remained somewhat unclear, whether this transfer would also hold when the CNN is not a more powerful teacher model.

Strengths include:
* The idea of regularizing the global token’s attention maps with the CNN activation maps is novel and interesting.
* The reported experimental improvements in the low-data regime are interesting.

Despite recommending the paper for rejection in its current form, I would like to encourage the authors to continue this line of work and present it again to the community with more focused discussions, insights (and possibly experiments). This is an interesting paper and it was evaluated to be close to (but below) the acceptance threshold.

**Award:**

No

---

### Decision · Program_Chairs · 2022-09-14

Reject